# Notes from the Underground: Heme Homeostasis in *C. elegans*

**DOI:** 10.3390/biom13071149

**Published:** 2023-07-19

**Authors:** Caiyong Chen, Iqbal Hamza

**Affiliations:** 1MOE Key Laboratory of Biosystems Homeostasis and Protection, College of Life Sciences, Zhejiang University, Hangzhou 310058, China; 2Center for Blood Oxygen Transport and Hemostasis, Department of Pediatrics, School of Medicine, University of Maryland, Baltimore, MD 21201, USA; 3Department of Animal and Avian Sciences, University of Maryland, College Park, MD 20742, USA

**Keywords:** heme, porphyrin, iron, *C. elegans*, trafficking, homeostasis

## Abstract

Heme is an iron-containing tetrapyrrole that plays a critical role in various biological processes, including oxygen transport, electron transport, signal transduction, and catalysis. However, free heme is hydrophobic and potentially toxic to cells. Organisms have evolved specific pathways to safely transport this essential but toxic macrocycle within and between cells. The bacterivorous soil-dwelling nematode *Caenorhabditis elegans* is a powerful animal model for studying heme-trafficking pathways, as it lacks the ability to synthesize heme but instead relies on specialized trafficking pathways to acquire, distribute, and utilize heme. Over the past 15 years, studies on this microscopic animal have led to the identification of a number of heme-trafficking proteins, with corresponding functional homologs in vertebrates. In this review, we provide a comprehensive overview of the heme-trafficking proteins identified in *C. elegans* and their corresponding homologs in related organisms.

## 1. Introduction

Heme, an iron-containing porphyrin, is an essential macrocycle for virtually all living organisms. It serves as a cofactor for a variety of proteins, such as globins, cytochromes, cytochrome P450s, catalases, and peroxidases [1,2,3]. Heme is also involved in biological processes, such as signal transduction, gene expression, circadian rhythm, and microRNA processing [4,5,6,7,8,9,10,11,12,13,14,15]. However, due to its intrinsic hydrophobic and pro-oxidant properties, free heme can intercalate into membrane lipids and cause cellular damage [2,16,17]. Therefore, organisms require specific transporters and chaperones to safely deliver heme from the sites of synthesis or uptake to other cellular destinations for incorporation into hemoproteins. Over the past 20 years, researchers have identified a growing number of heme-trafficking proteins that regulate heme homeostasis in eukaryotes [2,3]. 

*Caenorhabditis elegans* is a free-living roundworm widely used as a model organism in biological research. Although the adult worm is only ~1 mm in length, this animal has multiple tissues, including pharynx, intestine, neurons, muscle, hypodermis, cuticle, and reproductive tissues [18]. While the vast majority of animals can synthesize heme by using the substrates ferrous iron, glycine, and succinyl-CoA, the roundworm *C. elegans* lacks the heme biosynthetic pathway, despite containing homologs for numerous hemoproteins [19]. The *C. elegans* genome contains homologs for 33 globins [20] and 76 cytochromes P450s [21], as well as many genes for respiratory cytochromes, peroxidases, catalases, and soluble guanylate cyclases [22]. As a heme auxotroph, *C. elegans* has to take up heme from its food via the intestinal cells and distribute it to extra-intestinal tissues. Therefore, *C. elegans* is a unique animal model for studying heme transport pathways, as all endogenous heme is derived from dietary sources. Transcriptomics and functional genomics studies in this model organism have led to the identification of a number of proteins that play important roles in the import, export, intracellular and intercellular transport, and inter-tissue signaling of heme [23,24,25,26,27,28]. In this review, we will discuss the current understanding of heme-trafficking proteins in *C. elegans,* as well as the conserved roles of these pathways in other organisms. 

## 2. Heme Import

Because there is no endogenous heme production, all heme in the worm is provided by intestinal absorption. The *heme-responsive gene-4* (*hrg-4*) encodes a heme importer, with four transmembrane regions, that is critical for dietary heme acquisition in *C. elegans* [23,29] (Figure 1). When environmental heme is low, *hrg-4* is highly upregulated in worm intestinal cells, and the protein localizes to the apical surface. Knockdown of *hrg-4* leads to diminished heme assimilation, as revealed by the reduced uptake of the fluorescent heme analog zinc mesoporphyrin (ZnMP), resistance to the toxic heme analog gallium protoporphyrin IX (GaPP), and the heme deficiency response exhibited in the IQ6011 heme sensor worm, which carries a GFP reporter driven by the *heme-responsive gene-1*, *hrg-1* promoter [23,29]. The heme transport activity of HRG-4 was further confirmed by in vitro electrophysiological assays in Xenopus oocytes [23]. Structure–function analysis of HRG-4 by ectopically expressing it in a heme-deficient *Saccharomyces cerevisiae* strain further revealed that a tyrosine in the second transmembrane region, a histidine in the second exoplasmic loop, and a FARKY motif in the cytoplasmic C-terminus were critical for the heme transport activity [30].

Another *C. elegans* gene that is induced by low heme, *hrg-2*, has been implicated in heme utilization by the hypodermis [25] (Figure 1). HRG-2 is a type I membrane protein with thioredoxin-like and glutathione S-transferase (GST) domains. It localizes to the endoplasmic reticulum and apical plasma membrane in worm hypodermal cells [25]. Loss of *hrg-2* leads to aberrant cytochrome heme profiles, whereas heterologous expression of *hrg-2* in the *hem1Δ* yeast improves growth and oxygen consumption at submicromolar concentrations of exogenous heme [25]. The HRG-2 homolog in the barber’s pole worm *Haemonchus contortus*, Hc-HRG-2, displays activities of both heme binding and GST [31]. Several other GSTs, including GST-1, GST-2, and GST-3 in the hookworm *Necator americanus* and GST-19 in *C. elegans*, have also been implicated in heme transport and detoxification [32,33]. However, HRG-2 and these GSTs are unlikely to be heme transporters, as they do not contain multiple transmembrane domains [25,31,33]. Instead, they may regulate heme utilization indirectly by coordinating with other heme transporters or catalyzing enzymatic reactions. 

## 3. Heme Storage and Mobilization

In *C. elegans*, a proportion of heme is stored in the intestine. Studies using ZnMP as the heme tracer suggested that heme is concentrated in granular structures within *C. elegans* intestinal cells [19]. In vivo imaging of heme by high-resolution transient absorption microscopy demonstrated that these heme granules are indeed lysosomal-related organelles (LROs), as the heme signal colocalizes with the autofluorescence of gut granules, as well as the LRO marker GLO-1::GFP [34]. Knockdown of *hrg-4* led to diminished signals in both ZnMP and heme in LROs [19,34], indicating that the stored heme is derived, at least in part, via the HRG-4-mediated pathway. The precise mechanism responsible for the heme deposition into LROs still remains unclear. It is possible that a fraction of dietary heme enters LROs via the endocytic pathway. Indeed, in the fission yeast *Schizosaccharomyces pombe*, imported heme analog ZnMP was first observed in the storage site, vacuoles, prior to its appearance in the cytoplasm [35]. During this process, heme first interacts with a cell-surface-anchored protein called Shu1 and then undergoes internalization via the endocytic pathway [35,36].

In *C. elegans*, the stored heme can be mobilized out of LROs into the cytoplasm by the HRG-4 paralog HRG-1, a heme transporter that primarily localizes on LRO membranes [23] (Figure 1). The expression of *hrg-1* is upregulated under heme-limiting conditions [23]. RNAi knockdown of *hrg-1* causes ZnMP accumulation in LROs, indicative of the impaired mobilization of stored heme [23]. Consistent with its intracellular localization, HRG-1 preferentially interacts with heme at an acidic pH [23]. The *C. elegans* genome contains four *hrg-1* homologs. While *hrg-1* and *hrg-4* are upregulated under low heme, the other two genes, *hrg-5* and *hrg-6*, do not show transcriptional regulation by heme [30]. HRG-1 requires a histidine residue in the second exoplasmic loop and a FARKY motif at the C-terminal region for transporting heme [30]. Unlike HRG-4 and its paralog HRG-6, HRG-1 has histidine instead of tyrosine in the second transmembrane domain [23,30]. Since tyrosine is known to interact with oxidized heme [37,38,39], this difference implies that HRG-1 and HRG-4 may encounter different heme oxidation states. HRG-5 has a histidine residue at a slightly different position within the same transmembrane domain, which might be involved in heme transport [23]. The *hrg-1* orthologs in the filarial nematode *Brugia malayi* and barber’s pole worm *Haemonchus contortus*, *BmHRG-1* and *HcHRG-1*, are also regulated by heme levels [40,41]. BmHRG-1 and HcHRG-1 localize to both the cell surface and endocytic compartments, suggesting that they may have the function of both *C. elegans* HRG-1 and HRG-4 [40,41]. 

An intestinal-enriched transcriptomics analysis revealed two new HRGs, HRG-9 and HRG-10, that play important roles in mobilizing heme out of LROs [28] (Figure 1). Intestines were isolated from adult worms that had been cultured with varying concentrations of heme and subjected to transcriptomic analyses using the SMART-seq technology [42]. *hrg-9* was one of the genes induced by low heme in worm intestinal cells, while the expression of its paralog, *hrg-10*, was not regulated by heme [28]. Depletions of *hrg-9* and *hrg-10* induced heme deficiency responses in the heme sensor strain carrying the *hrg-1p::gfp* heme sensor reporter [28]. Consistently, the *hrg-9* and *hrg-10* knockout worms displayed reduced sensitivity to GaPP toxicity, even though neither total heme nor heme uptake were altered in the *hrg-9* and *hrg-10* mutants [28]. Results from ZnMP assays indicate that the knockout of *hrg-9* and *hrg-10* leads to heme accumulation in LROs, a defect that can be further exacerbated by the knockdown of *hrg-1* [28]. Because HRG-9 and HRG-10 do not have predicted transmembrane regions, these observations suggest that these two proteins may serve as chaperones to deliver heme from LROs.

## 4. Transport of Heme to Other Tissues

Given that *C. elegans* cannot make heme, extraintestinal tissues such as neurons, muscles, and hypodermis have to acquire heme from the intestine. Therefore, intestinal heme must be exported into the worm’s body cavity, the pseudocoelom, for delivery to other tissues. Multidrug resistance protein-5 (MRP-5) or ABCC5, an ATP-binding cassette transporter with 12 transmembrane domains, plays a critical role in translocating heme from the intestine into the circulation [26] (Figure 1). In the intestinal cells, MRP-5 mainly localizes to basolateral membranes [26]. Its depletion leads to embryonic lethality and growth arrest, phenotypes that can be rescued by heme supplementation [26]. Consistent with its role as a major intestinal heme exporter, *mrp-5*-deficient worms show elevated levels of ZnMP and heme in the intestine, resistance to the toxic heme analog GaPP, and a concomitant deficiency in extra-intestinal heme levels [26,29,43]. Expression of *C. elegans mrp-5* in yeast leads to enhanced heme loading into the secretory pathway, providing further evidence to support a role for MRP-5 in exporting cytosolic heme into the lumen [26]. Interestingly, other tissues, including pharynx, hypodermis, and neurons, also express *mrp*-5, indicating that the function of MRP-5 is not limited to the intestine in the worm [26].

During heme starvation, *C. elegans* expresses *hrg-3*, another heme-responsive gene, for supplying heme to developing oocytes [24] (Figure 1). HRG-3 is a small heme-binding protein with a signal peptide at the N-terminus. Biochemical analyses indicate that heme is coordinated with the conserved histidine residues and a hydrophobic core formed in the HRG-3 dimer [44]. *hrg-3* is predominantly expressed in worm intestinal cells under heme deficiency [24]. The HRG-3 protein, likely in complex with heme, is loaded into secretory vesicles and secreted into the circulation, where it can further traffic to worm reproductive tissues [24]. The knockout of *hrg-3* in the worm does not induce overt phenotypes under standard growth conditions [24]. However, when *hrg-3* knockout worms are cultured on a low-heme diet, their progeny either die during embryogenesis or are arrested immediately following hatching [24]. These defects are suppressed by the maternal but not zygotic expression of *hrg-3* [24]. These results suggest an important role for HRG-3 in transferring heme from the maternal intestine to developing oocytes. In *C. elegans*, the destination of the secreted HRG-3 may not be restricted to the germline because it is also expressed during embryonic and larval stages and in males [24]. 

## 5. Regulation of Heme-Responsive Genes in *C. elegans*

In mammals, heme is known to interact with and regulate a number of transcription factors, including Bach1, Rev-erbs, NPAS2, PER2, and p53 [6,9,10,11,12,45]. Heme binding can stimulate the nuclear export and degradation of Bach1 and p53 [7,8,9,46]. Additionally, heme may regulate the interactions between the transcription factors and DNA, as well as other trans-acting factors [6,9,10,12,45,47]. 

Microarray analyses performed on *C. elegans* revealed 288 heme-responsive genes, including several aforementioned *hrgs*, that displayed differential expressions in response to varying heme levels [29]. These genes were implicated in a variety of cellular processes, such as lipid metabolism, electron transport, proteolysis, development, and reproduction [29]. An intestinal-enriched RNA-seq experiment further showed that heme might regulate the expression of over 500 genes in *C. elegans* intestinal tissues [28]. Transgenic analyses with transcriptional fusion reporters suggest that many known *hrgs,* including *hrg-1*, *hrg-2*, *hrg-3*, *hrg-4*, *hrg-7*, and *hrg-9,* are regulated by heme at the transcriptional level [23,24,25,27,28]. Since the expressions of these *hrg* reporters negatively correlate with the heme levels, some of them have been used as sensors to monitor the heme status in the worm [23,26,27,28]. A genome-wide RNAi study identified 177 genes that regulate the expression of the *hrg-1* reporter, over 30 of which are components of the mitochondrial electron transport chain or ATP synthase [27]. Since heme is an essential electron carrier in the respiratory chain, this regulation implies a feedback response to mitochondrial dysfunction by modulating heme homeostasis. 

The expression of *hrg-1* can also be regulated by inter-organ signaling between the intestine and extra-intestinal tissues [27]. When heme is limiting, the intestine expresses and secretes a signaling factor called HRG-7, a cathepsin protease homolog that localizes to anterior and posterior sensory neurons. Conversely, the *C. elegans* neurons secrete DBL-1, a homolog of bone morphogenetic protein 5, to repress the expressions of *hrg-7* and *hrg-1* in the intestine through the transcription factor SMA-9 [27]. This inter-organ signaling provides a mechanism for coordinating the intestinal heme absorption with organismal heme status. Loss of *hrg-7,* as well as *dbl-1* and *sma-9,* perturbs the intestinal response to systemic heme deficiency, as demonstrated by the altered expressions of *hrg-1* and *hrg-7* [27]. HRG-7 shows homology to the A1 family of aspartic proteases and contains two conserved aspartate residues in the active site, but mutations of these two residues do not affect its function in heme signaling [27]. Thus, HRG-7 may regulate inter-organ signaling through a mechanism that is independent of the aspartate protease activity. 

While the majority of the characterized *hrgs* are upregulated by heme starvation, their heme-dependent expressions may be controlled by distinct transcriptional mechanisms. Truncation and mutagenesis analyses revealed a 23-bp heme-responsive element (HERE) in the *hrg-1* promoter that was critical for the transcriptional regulation by heme [48]. This cis element may coordinate with nearby GATA elements, the binding sites for ELT-2, to drive the intestinal expression of *hrg-1* in a heme-dependent manner [48]. The HERE is also present in the promoter regions of *mrp-5* and *hrg-7* [48]. However, the promoters of several other *hrgs,* such as *hrg-2*, *hrg-3*, and *hrg-4,* do not contain this element, suggesting that these genes may be regulated by heme through other mechanisms. Currently, the transcription factors and the precise mechanisms responsible for the heme-dependent regulation of *hrg-1,* as well as other *hrgs,* are unknown. 

## 6. Homologs of *C. elegans* Heme-Trafficking Proteins in Other Organisms

The research on *C. elegans* has significantly promoted the understanding of heme transport pathways in vertebrates and other organisms. Several heme-trafficking genes, including *hrg*-1, *hrg*-4, *mrp*-5, and *hrg*-9, have homologs with conserved roles in other eukaryotes. 

### 6.1. HRG-1 Homologs

Orthologs of *hrg-1* have been identified and characterized in mammals, fish, ticks, parasitic nematodes, and trypanosomatids [23,40,41,49,50,51,52,53,54,55]. In mammals, the vast majority of iron used for the synthesis of heme and hemoglobin in differentiating erythroblasts is supplied by tissue macrophages, which are responsible for recycling iron from senescent red blood cells (RBCs) [56,57]. The mammalian *HRG1* (S*LC48A1*) is highly expressed during erythrophagocytosis, a process in which macrophages engulf and destruct aged RBCs, and the protein localizes preferentially to erythrophagosomal membranes [49]. HRG1 transports heme derived from degraded RBCs inside erythrophagosomes to the cytosol, and subsequently, heme is degraded by heme oxygenase to release iron or is exported out of the cell [49]. Knockout *Hrg1* in mice leads to impaired erythrophagocytosis, with over 10-fold excess heme in the lysosomes of reticuloendothelial macrophages [51]. The heme accumulates as hemozoin [51], heme crystals previously found only in blood-feeding organisms [58,59]. As a consequence of impaired heme recycling, the *Hrg1* knockout mice are susceptible to dietary iron deficiency [51]. 

In zebrafish, the systemic heme recycling from senescent RBCs takes place in the kidney [50], which is also the hematopoietic site at larval and adult stages [60]. The zebrafish genome contains two *hrg1* homologs, *hrg1a* (*slc48a1b*) and *hrg1b* (*slc48a1a*), which are also required for recycling heme from senescent RBCs [50]. Zebrafish lacking both *hrg1a* and *hrg1b* accumulated a much higher amount of heme in kidney macrophages than wild-type controls during acute hemolysis induced by phenylhydrazine [50]. Furthermore, the loss of *hrg1* in zebrafish and mice caused the aberrant expression of iron and heme metabolism genes [50,51]. Heterologous expressions of mammalian *HRG1* and fish *hrg1* promoted the growth of *hem1Δ* yeast under heme-limiting conditions, confirming their function in heme transport [30,50]. Taken together, the HRG1 homologs in vertebrates play an important role in recycling heme iron from macrophages of the reticuloendothelial system [49,51] (Figure 2). It is noteworthy to mention that *HRG1* is also expressed in other tissues, such as brain, heart, skeletal muscle, intestine, and lung [23], implying a broader role for HRG1 in heme trafficking in those organs. 

The role of HRG1 in intracellular heme transport is conserved in another heme auxotrophic organism, the arthropod *Ixodes ricinus* [52,61]. Although the expression of the *I. ricinus hrg-1* homolog, *IrHRG,* is not responsive to the dietary hemoglobin level, its silencing leads to reduced toxicity to GaPP and accumulated heme in hemosomes of midgut digestive cells [52]. The heme transport activity of IrHRG is further verified by its ability to rescue the growth of the *hem1Δ* yeast [52]. These observations suggest that IrHRG mediates the transport of heme released from digested host hemoglobin out of hemosomes (Figure 2).

Trypanosomatid parasites are a family of protozoans that are also unable to synthesize heme because they lack several enzymes in the heme biosynthetic pathway [62,63]. The HRG-1 family protein in *Leishmania amazonensis*, Leishmania Heme Response-1 (LHR1), has been shown to be critical for heme uptake [53] (Figure 2). Similar to *C. elegans hrg-1* and *hrg-4*, *LHR1* is a heme-responsive gene that is upregulated by low heme, and the protein localizes to the plasma membrane and lysosomal compartments [53]. Complete ablation of *LHR1* is lethal to *L. amazonensis*, while the deletion of one allele reduces ZnMP uptake and intracellular heme levels [53]. Importantly, the LHR1-mediated heme uptake is critical for both the virulence and survival of *L. amazonensis* [54,55]. The LHR1 homolog in another trypanosome species, *Trypanosoma cruzi,* has been demonstrated to transport heme as well [64,65] (Figure 2), implying that LHR1 plays a conserved role in heme assimilation in trypanosomes. While the conserved histidine residues and FARKY motif are critical for heme transport in most metazoan HRG-1s [23,30,49,52,66], the trypanosomal HRG-1 homologs appear to use a different set of residues, including tyrosine residues in the first, third, and fourth transmembrane regions, to coordinate heme transfer [55].

### 6.2. MRP-5 Homologs

MRP5 belongs to the ATP-binding cassette transporter subfamily C. A functional analysis of MRP family genes in *Drosophila melanogaster* demonstrates that the fly MRP5, called dMRP5 or CG4562, regulates heme homeostasis [67] (Figure 2). Heme treatment significantly enhances the expression of *dMRP5* in Schneider 2 (S2) cells, as well as in the fly gut [67]. Silencing of *dMRP5* leads to heme accumulation in the intestine and animal lethality, which can be suppressed by the heme synthesis inhibitor, succinylacetone, or by overexpressing the fly heme oxygenase gene [67]. Additionally, overexpression of *dMRP5* leads to reduced ZnMP levels in S2 cells [67]. These data consistently indicate that dMRP5 functions as a heme exporter in fruit flies [67]. 

Humans have nine *MRP* genes [68], among which *MRP5* shows the highest homology to *C. elegans mrp-5*. MRP5 displays distinct localization patterns in different types of mammalian cells. In polarized MDCKII, an epithelial-like cell line, MRP5 mainly localizes to the basolateral membrane [26]. In mouse embryonic fibroblasts (MEFs) and mouse testes, MRP5 is enriched in intracellular vesicles and mitochondrial-associated membranes, respectively [26,69]. The losses of *MRP5* and its closely related paralog *MRP9* in mice result in mitochondrial dysfunction in testes, which leads to reproductive defects [69]. *MRP5*-deficient MEFs exhibit reduced heme incorporation into the secretory pathway, supporting a role for MRP5 in heme export [26] (Figure 2). Since MRP5 proteins have also been implicated in transporting a variety of other substrates, such as cyclic nucleotides, folate, hyaluronan, glutamate conjugates, and vitamin B12 [70,71,72,73,74], further investigations are needed to determine the physiological substrate of MRP5 in mammals.

### 6.3. HRG-9 Homologs

HRG-9 is homologous to the transport and Golgi organization 2 (TANGO2) group of proteins. *TANGO2* was originally identified as one of the genes that regulates protein secretion and Golgi organization in a genome-wide RNAi screen in *Drosophila* S2 cells [75]. In mammalian cells, TANGO2 mainly localizes to the cytoplasm and mitochondria [28,76]. Knockout of *TANGO2* causes mitochondrial heme accumulation in both human embryonic kidney (HEK293) cells and mouse erythroleukemia (MEL) cells [28]. Consistently, studies with genetically encoded fluorescent heme sensors in the budding yeast *S. cerevisiae* revealed that the deletion of *tango2* led to heme accumulation in the mitochondria and decreased heme content in the cytosol [28,77]. Biochemical assays further demonstrated that TANGO2 is a low-affinity heme-binding protein that is able to transfer heme from the mitochondria [28]. These observations suggest that TANGO2 is a cytosolic heme chaperone that transports heme from heme-enriched compartments, such as the mitochondria (Figure 2). As neither HRG-9 nor TANGO2 contains transmembrane domains, it is likely that they can extract heme from heme-enriched membranes and deliver it down a concentration gradient. In this case, heme has to be translocated from the mitochondrial matrix, where the newly synthesized heme is released, to the cytosolic leaflets of mitochondrial membranes. This translocation may require other mitochondrial heme-trafficking proteins, such as the feline leukemia virus subgroup C receptor 1b (FLVCR1b), progesterone receptor membrane component 1 (PGRMC1), or PGRMC2 [78,79,80]. Currently, the precise mechanism underlying the TANGO2-mediated heme extraction from membrane lipids remains to be investigated. 

Mutations in human *TANGO2* can cause an inherited disease, which exhibits pleiotropic symptoms, including developmental delay, rhabdomyolysis, arrhythmias, encephalopathy, and metabolic crisis [81,82,83]. Correspondingly, loss of *tango2* in zebrafish leads to arrhythmia, myopathy, encephalopathy, and death during early development [28]. The symptoms induced by *TANGO2* deficiency in humans, as well as in flies, can be alleviated by supplementing with B vitamins [83,84], indicating that these symptoms may be associated with defective metabolism. At the cellular level, defects in mitochondrial function and ER-to-Golgi trafficking have been observed in *TANGO2*-deficient cells [28,76,81,82,85,86,87]. Further studies are required to elucidate how the defects in heme homeostasis, mitochondria metabolism, or membrane trafficking induce the clinical symptoms during *TANGO2* deficiency. 

Besides eukaryotes, many bacteria, archaea, and some viruses have putative homologs of *hrg-9*/*TANGO2* [88]. The homologous gene in the Gram-negative γ-proteobacteria *Shewanella oneidensis* was also shown to encode a heme chaperone, which was named heme-trafficking protein A (HtpA) [88]. *S. oneidensis* has 42 genes for cytochrome *c* [89] and thus has a high demand for heme transport during cytochrome *c* maturation (CCM). HtpA binds heme with a 1:1 stoichiometry and a *K*_d_ of ~1.2 μM [88]. Overexpression of *HtpA* increases the cytochrome *c* content in both the wild-type *S. oneidensis* and the *ccmI* mutant, a strain that is defective in CCM [88]. HtpA may facilitate heme transfer to the CCM system by interacting with CcmB at the cytoplasmic side [88]. Loss of HtpA also results in the reduced activity of the heme-containing enzyme catalase, indicating that bacterial HtpA may play a broader role in heme trafficking [88].

Apart from the progresses made in heme homeostasis by studying the model organism *C. elegans*, a number of other heme-trafficking proteins have been identified in mammals and other organisms. FLVCR1a is a heme exporter with critical roles in recycling heme from macrophages that ingest senescent red blood cells and exporting excess heme in developing erythroblasts to maintain the balance between the heme availability and globin synthesis [90,91,92,93]. A short FLVCR1a variant named FLVCR1b may export heme out of mitochondria [78]. Another major facilitator superfamily protein, FLVCR2, and its putative homolog, LmFLVCRb, were reported to be heme importers in mammalian cells and *Leishmania* parasites, respectively [94,95]. PGRMC1 and PGRMC2 may interact with ferrochelatase, the terminal heme synthesis enzyme, to facilitate heme transport from the mitochondria to the endoplasmic reticulum and nucleus [79,80]. In addition, proteins with previously known functions, such as glyceraldehyde phosphate dehydrogenase and heat shock protein 90, have recently been implicated in buffering cellular heme and inserting it into downstream hemoproteins [96,97,98,99,100].

## 7. Conclusions

The last two decades have witnessed a tremendous growth in the understanding of heme-trafficking pathways in eukaryotes. Many of the important insights were gained by studying *C. elegans*. This roundworm, as well as all other animals in the phylum Nematoda, lacks the heme biosynthetic pathway and thus completely depends on trafficking pathways to acquire and transport heme. Transcriptomics and genome-wide RNAi studies in *C. elegans* have uncovered a number of proteins, most of which are named HRGs, that play important roles in the uptake, export, utilization, intercellular transport, and signaling of heme. The functions of several HRGs, including HRG-1, HRG-4, HRG-9, and MRP-5, are conserved in vertebrates and other organisms. In the future, the application of forward genetic screens, newly engineered heme sensors, and other genetic tools to the heme-trafficking field will further advance our understanding of heme homeostasis in metazoans. 

## Figures and Tables

**Figure 1 biomolecules-13-01149-f001:**
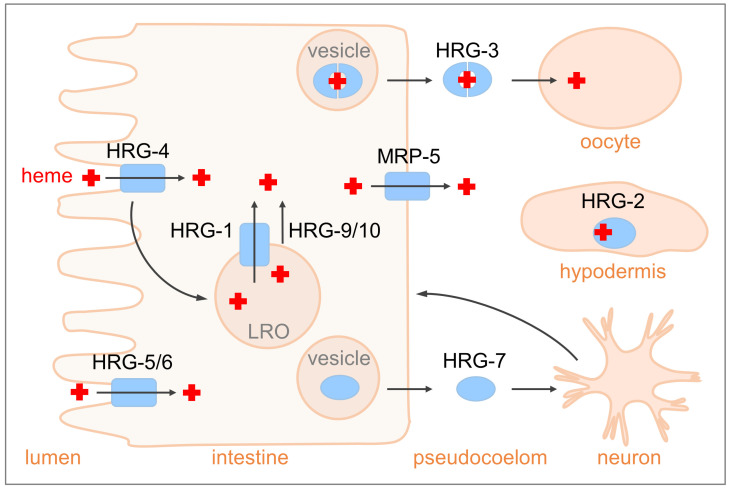
Heme-trafficking pathways in *C. elegans*. *C. elegans* is a heme auxotroph and thus requires trafficking proteins to import, allocate, export, and utilize heme. The heme importer HRG-4 and its paralogs HRG-5 and HRG-6 take up heme from the gut lumen into intestinal cells. The imported heme may be stored in lysosomal-related organelles (LROs) within intestinal cells. Another HRG-4 paralog, HRG-1, and the heme chaperones HRG-9 and HRG-10 are responsible for mobilizing heme out of LROs. The heme exporter MRP-5 transports heme across basolateral membranes of intestinal cells into the pseudocoelom, a body cavity filled with fluid, from where heme is delivered to other tissues, including neurons, muscles, hypodermis, and embryos. HRG-3 is a secreted heme-binding protein that facilitates heme transfer from the maternal intestine to developing oocytes. HRG-2 facilitates heme acquisition or utilization in the hypodermis. HRG-7 is a secreted protein that transmits heme-starvation signals to neurons, which in turn regulate systemic heme homeostasis by modulating the expression of heme-trafficking genes in intestinal cells.

**Figure 2 biomolecules-13-01149-f002:**
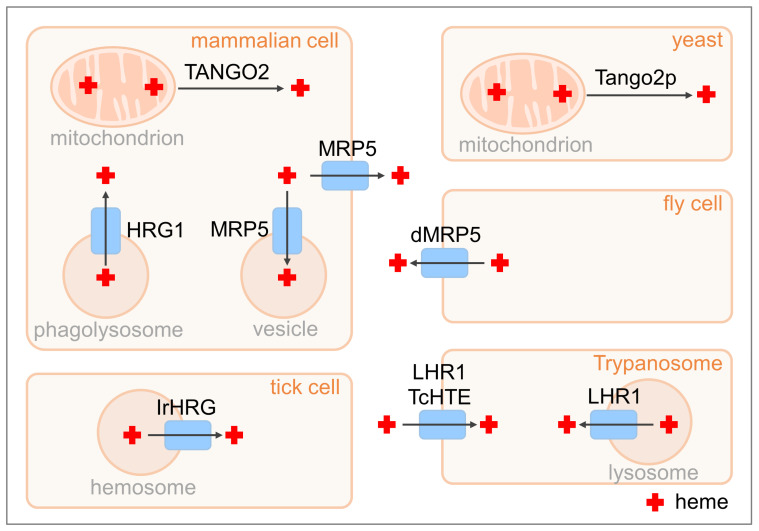
Conserved roles for HRG homologs in non-nematode organisms. The HRG-9 ortholog TANGO2 transfers heme out of the mitochondria in mammalian cells and yeast. Mammalian MRP5 may transport heme into the secretory vesicles or across the basolateral membranes, depending on the cell type. The *Drosophila* MRP5 (dMRP5) mediates heme transport out of intestinal cells. In macrophages of the mammalian reticuloendothelial system (spleen, liver, and bone marrow), HRG1 transports heme from the erythrophagosomes into the cytosol. The HRG-1 homolog in the hard tick *I. ricinus*, IrHRG, transports heme released from digested host hemoglobin out of hemosomes. The HRG-4 homologs LHR1 and TcHTE are heme importers in trypanosomatid parasites *L. amazonensis* and *T. cruzi*, respectively. LHR1 may also be involved in mobilizing heme out of lysosomal compartments.

## Data Availability

Not applicable.

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
