# Peer review of "Notes from the Underground: Heme Homeostasis in *C. elegans"

_biomolecules, 2023, doi:10.3390/biom13071149_

Round 1

Reviewer 1 Report

Heme is an essential iron-containing macrocycle whose transport must be carefully coordinated due to its toxic nature. The nematode C. elegans has proved to be a pioneering organism in the discovery/characterization of novel heme trafficking proteins. This has been empowered by the fact that C. elegans is a heme auxotroph totally dependent upon exogenous heme supply. In this review, Notes from the underground: Heme homeostasis in C. elegans, Chen and Hamza provide a comprehensive overview of the heme trafficking proteins that have been identified in C. elegans over the past ~15 years. They also provide meaningful connections between newly discovered C. elegans proteins and their homologs in other organisms, including mammals.  

This review is thorough, appropriately cited, and a meaningful addition to the literature in the field of heme biology and transport. Below is a list of minor issues that the authors should consider during revision. The primary issue that the authors should consider is improving Figure 2 which is oversimplified and does not contain the information content that is present in Figure 1. 

Issues:

1: Figure 2 is quite complicated and an oversimplification, understandable given that the authors are using a single cell to model multiple species and cell types. For instance, why have the authors chosen to place MRP5 on an unlabeled intracellular vesicle? In section 6.2, lines 214-217, the authors point out that mammalian MRP5 has multiple localization patterns depending upon cell type (basolateral membrane, intracellular vesicles, mitochondrial-associated membranes). I appreciate that including all described localization patterns would possibly clutter the figure. But, given that this is a comprehensive review, I think the authors should strive to make Figure 2 more complete. To make Figure 2 a more complete synopsis of homologous pathways in non-C. elegans organisms, the authors should consider adding labels to different compartments inside the cell. Also, a more complete figure legend might help provide additional context. One example: line 192, the sentence “MRP5 may serve as a heme exporter in mammalian cells and the fruit fly” should be expanded to describe which mammalian MRP5 is being presented and where it is localized. In contrast, Figure 1 is a very thorough and palatable graphical synthesis and could be used as an intellectual template to improve Figure 2.

2: In line 48, the authors reference a “heme sensor worm [22,28].” It would be useful to define this reagent and its origins here as this is its first mention in this review. I believe this same strain is described/referenced later in line 100. 

3: In the description of HRG-4 structure-function analysis (lines 49-51), the authors describe the FARKY motif and additional critical residues characterized by assays in a heterologous yeast system. I believe it would be worthwhile for the authors to comment on the evolutionary novelty of these residues/motifs. Are these common among all the C. elegans HRG-4 homologs (HRG-1, HRG-5, HRG-6)? Are these features found in other heme-trafficking proteins that are not homologous to HRG-4? Are these features/motifs found in other species, including those mentioned in section 6.1 “HRG-1 homologs”?

Typos:

4: There is an extra space after the citation in line 31. “[2,3] .”

5: In line 67 the authors use the phrase “circulation pseudocoelom”. This word choice seems incorrect, consider “circulating pseudocoelomic fluid.”

6: In line 74, the authors use the plural “intestines” when the singular “intestine” seems more appropriate.

Author Response

General comments: 

“This review is thorough, appropriately cited, and a meaningful addition to the literature in the field of heme biology and transport. Below is a list of minor issues that the authors should consider during revision. The primary issue that the authors should consider is improving Figure 2 which is oversimplified and does not contain the information content that is present in Figure 1.” 

We thank the reviewer for the comment. We have modified Figure 2 as requested. Please refer to the following response to Comment #1 for details. 

Minor comments:

 1 “Figure 2 is quite complicated and an oversimplification, understandable given that the authors are using a single cell to model multiple species and cell types. For instance, why have the authors chosen to place MRP5 on an unlabeled intracellular vesicle? In section 6.2, lines 214-217, the authors point out that mammalian MRP5 has multiple localization patterns depending upon cell type (basolateral membrane, intracellular vesicles, mitochondrial-associated membranes). I appreciate that including all described localization patterns would possibly clutter the figure. But, given that this is a comprehensive review, I think the authors should strive to make Figure 2 more complete. To make Figure 2 a more complete synopsis of homologous pathways in non-C. elegans organisms, the authors should consider adding labels to different compartments inside the cell. Also, a more complete figure legend might help provide additional context. One example: line 192, the sentence “MRP5 may serve as a heme exporter in mammalian cells and the fruit fly” should be expanded to describe which mammalian MRP5 is being presented and where it is localized. In contrast, Figure 1 is a very thorough and palatable graphical synthesis and could be used as an intellectual template to improve Figure 2. 

These are excellent suggestions. In the revised manuscript, we have modified Figure 2 extensively by using different cell models for different species. We have also added the labels for all the intracellular compartments and expanded the figure legends accordingly. 

2 “In line 48, the authors reference a “heme sensor worm [22,28].” It would be useful to define this reagent and its origins here as this is its first mention in this review. I believe this same strain is described/referenced later in line 100. 

We have added a description of this reporter strain in the text.

 3 “In the description of HRG-4 structure-function analysis (lines 49-51), the authors describe the FARKY motif and additional critical residues characterized by assays in a heterologous yeast system. I believe it would be worthwhile for the authors to comment on the evolutionary novelty of these residues/motifs. Are these common among all the C. elegans HRG-4 homologs (HRG-1, HRG-5, HRG-6)? Are these features found in other heme-trafficking proteins that are not homologous to HRG-4? Are these features/motifs found in other species, including those mentioned in section 6.1 “HRG-1 homologs”?

 This is an excellent question regarding the function of HRG-1 family members. Histidine and tyrosine are common heme-binding residues with important roles in heme transport. In response to the reviewer’s comment, we discuss these functional features of HRG-1 homologs in Section 3 and Section 6.1 in the revised manuscript. The function of these residues has not been well-characterized in HRG-1 homologs from other species.

4 “There is an extra space after the citation in line 31. “[2,3] .”

 We have deleted the extra space.

5 “In line 67 the authors use the phrase “circulation pseudocoelom”. This word choice seems incorrect, consider “circulating pseudocoelomic fluid.”

 We thank the reviewer for the suggestion. We have made the change in the figure legend.

 6 “In line 74, the authors use the plural “intestines” when the singular “intestine” seems more appropriate.

 We have made the requested changes to the text.

Reviewer 2 Report

This manuscript summarizes proteins involved in heme trafficking in C. elegans and their homologues. C. elegans is a good model for studying heme transport because it cannot synthesize heme on its own and uptakes heme from the environment. The author of this review is one of the top scientists in this field. References are properly cited. The text is well organized and very informative. The reviewer believes that this review will be of interest to many readers of Biomolecules and is therefore willing to accept this manuscript for publication after some trivial modifications.

Page 2, line 61. Add reference(s) to the last sentence of this paragraph.

Page 3, Figure 1. What do the circles with HRG-3 and HRG-7 present before secretion mean? Explanation is needed in the legend.

Author Response

Minor comments: 

  • “Page 2, line 61. Add reference(s) to the last sentence of this paragraph.

We have added references to the second to the last sentence of the paragraph. The last sentence is the authors’ viewpoint.

2 “Page 3, Figure 1. What do the circles with HRG-3 and HRG-7 present before secretion mean? Explanation is needed in the legend.

We thank the reviewer for the comment. These circles indicate secretory vesicles. We have now labeled all the intracellular compartments in both revised Figures.

Reviewer 3 Report

This review of heme-binding and trafficking in C.elegans and other animals is excellent and I enjoyed reading it.

I only have a few comments and remarks:

For the benefit of non-experts, the authors may add an introductory section briefly describing C.elegans anatomy and physiology. 

Line 27: ”other biological processes”, other is not quite appropriate since no biological processes are mentioned in preceding sentences. I suggest writing: “Heme is also involved in biological processes such as signal transduction….”

Line 119-128: How does secretion of HRG3 allow mobilization of heme from the intestinal cell? This should involve some kind of transport over the basolateral membrane. This could be discussed.

Author Response

 General comments:

 This review of heme-binding and trafficking in C.elegans and other animals is excellent and I enjoyed reading it. I only have a few comments and remarks. For the benefit of non-experts, the authors may add an introductory section briefly describing C.elegans anatomy and physiology.

We thank the reviewer for the suggestion. We have added a brief introduction of C. elegans to the beginning of the second paragraph. We have also included a reference for general information on C. elegans in case readers of this review are interested in learning more about worms.

 Minor comments:

 1 “Line 27: “other biological processes”, other is not quite appropriate since no biological processes are mentioned in preceding sentences. I suggest writing: “Heme is also involved in biological processes such as signal transduction….

We have updated the sentence according to the reviewer’s suggestion.

2 “Line 119-128: How does secretion of HRG3 allow mobilization of heme from the intestinal cell? This should involve some kind of transport over the basolateral membrane. This could be discussed.

 The HRG-3-heme complex is loaded into secretory vesicles and is secreted through the canonical secretory pathway across the basolateral membrane. We have modified the text to make this point clearer.